# Strong Modulation of Short Wind Waves and Ka-Band Radar Return Due to Internal Waves in the Presence of Surface Films. Theory and Experiment

**Stanislav A. Ermakov** [1,2,3], **Irina A. Sergievskaya** [1,2,3,*] and **Ivan A. Kapustin** [1,2,3]

[1] Institute of Applied Physics, Russian Academy of Sciences, 46 Uljanova Str., 603950 Nizhny Novgorod, Russia; stas.ermakov@ipfran.ru (S.A.E.); kia@ipfran.ru (I.A.K.)

[2] Faculty of Shipbuilding, Hydraulic Engineering and Environmental Protection, Volga State University of Water Transport, 5 Nesterova Str., 603950 Nizhny Novgorod, Russia

[3] Radiophysics Department, Lobachevsky State University of Nizhny Novgorod, 23 Gagarin Av., 603950 Nizhny Novgorod, Russia

[*] Correspondence: i.sergia@ipfran.ru; Tel.: +7-831-416-4935

**Abstract:** Strong variability of Ka-band radar backscattering from short wind waves on the surface of water covered with surfactant films in the presence of internal waves (IW) was studied in wave tank experiments. It has been demonstrated that modulation of Ka-band radar return due to IW strongly depends on the relationship between the phase velocity of IW and the velocity of drifting surfactant films. An effect of the strong increase in surfactant concentration was revealed in convergent zones, associated with IW orbital velocities in the presence of a "resonance" surface steady current, the velocity of which was close to the IW phase velocity. A phenomenological model of suppression and modulations in the spectrum of small-scale wind waves due to films and IW was elaborated. It has been shown that backscatter modulation could not be explained by the modulation of free (linear) millimeter-scale Bragg waves, but was associated with the modulation of bound (parasitic) capillary ripples generated by longer, cm–dm-scale waves—a "cascade" modulation mechanism. Theoretical analysis based on the developed model was found to be consistent with experiments. Field observations which qualitatively illustrated the effect of strong modulation of Ka-band radar backscatter due to IW in the presence of resonance drift of surfactant films are presented.

**Keywords:** Ka-band radar backscattering; internal waves; short wind waves; radar modulation transfer function; surfactant films

## 1. Introduction

It is well known that internal waves (IWs) play an important role in ocean dynamics, in particular in energy fluxes, mixing processes, the exchange of nutrients and gases, etc., and the detection and characterization of IW remains a challenging problem. Remote sensing methods, particularly using satellites, are the most promising techniques for investigations of IWs using IW signatures on the sea surface (see, e.g., [1–5] and references therein). Orbital surface currents caused by IWs modulate the spectrum of small-scale wind waves, thus resulting in significant variations of microwave radar return.

Different mechanisms by which IWs can be observed on the sea surface as bands with reduced and/or increased intensity of small-scale, gravity-capillary waves (GCWs) have been described in the literature. The most often discussed are hydrodynamic modulation (so-called straining or kinematic mechanism) of short wind waves due to non-uniform orbital velocities of the IW (see, e.g., [1,6–10]), and a "film modulation mechanism" associated with the redistribution of surfactants on the sea surface and with corresponding modulation of the surface wave damping rate (see, e.g., [11–14]. The straining mechanism is shown to be effective for those GCWs for which group velocities are close to the IW phase velocity—a resonance between IWs and surface waves. For typical oceanic IWs,

this mechanism can be realized for the dm-/m-scale wavelength range of the wind wave spectrum [8–10]. Short cm–mm-scale surface waves are far from the resonance with oceanic IW, and this mechanism is unable to explain observations in this wavelength range, in particular, radar signatures of oceanic IWs in X- and K-band radar images. Instead, cm–mm-scale waves are very sensitive to the presence of surface films, and variations in their spectrum can be rather significant when the surfactant concentration is modulated by non-uniform IW orbital currents. This mechanism was illustrated by some laboratory [11] and field observations [10,13] indicating the intensification of IW signatures in the presence of surfactant films, but rigorous quantitative explanations have not yet been given to this effect.

There has been another modulation mechanism—a "cascade" modulation associated with so-called bound waves (see, [15,16]), in particular with parasitic capillary ripples [10,17–21] generated by steep cm–dm-scale GCW when the latter are modulated, even if weakly, by IW. This "cascade" mechanism was proposed in [10] to explain the strong modulation of mm-scale parasitic ripples under the action of IW in a wave tank experiment and some of its manifestations in field observations. The capillary ripple modulations can be much stronger than that of the carrying steep short gravity waves because the intensity of the ripples grows very fast with the amplitude of the gravity waves. The experimental studies described in [10] were, however, mostly qualitative; thus, more detailed quantitative experiments and their theoretical analyses were needed. The difficulty of such analysis is some shortcomings of the model of the wind wave spectrum, because the sources, sinks and nonlinear effects affecting the formation of the wind wave spectrum are still poorly known. In [22,23], some phenomenological terms describing the generation of bound waves were introduced into the kinetic equation for the spectrum of the wind wave action. Bound waves in [22,23] were described as if they were quasi-linear and weakly interacting waves with their intrinsic dispersion relationship: this is not completely true. This model has difficulty explaining some phenomena observed in the experiments; in particular, an increase in capillary ripple modulation over internal and long surface waves in the presence of surfactant films [24,25]. Laboratory experiments [26] have indicated that parasitic capillary ripples excited on the forward slope of short steep gravity waves with frequencies of 3–7 Hz are stationary, i.e., "frozen" in the short gravity wave profile, sharply grow with the amplitude of the short gravity waves, and moved with the phase velocity of carrying short gravity waves. Therefore, despite the fact that the cascade mechanism announced in [10] was quite a long time ago and the dependencies of the ripple intensity on the slope of the short gravity waves were measured in a laboratory, a comprehensive analysis of the cascade mechanism has not been performed.

This paper aims to report on the results of laboratory/field studies of strong variability of capillary ripples due to IWs in the presence of surfactant films. A phenomenological model of the modulation of short wind waves and of Ka-band radar return due to IWs in the presence of surfactant films was developed and is presented in Section 2. Section 3 describes the procedures of wave tank experiments and, very briefly, of some field observations. Results on the strong modulation of the surfactant concentration and of Ka-band radar return in the IW orbital currents are presented in Section 4. Modulation of free Bragg ripples and parasitic capillary ripples due to the cascade mechanism are discussed in the context of the model in Section 5, and a comparison with experiments is presented in Section 5, as well. The obtained results are summarized in Section 6.

## 2. Theory

Here, we develop a model of modulation of small-scale, capillary surface waves with wavelengths of approximately 1 cm and shorter, and a model of radar backscattering in application for the analysis of experiments on the radar probing of wind waves in the presence of internal waves, surfactant films and a steady surface current.

For simplicity, we consider here only a one-dimensional case because the results will be further applied to the analysis of experiments in a narrow wind wave flume and

to field observations in which IW propagation, wind velocity and radar directions are nearly parallel.

### 2.1. Modulation of Surfactant Concentration Due to Internal Waves

The surfactant concentration $\widetilde{\Gamma} \equiv \Gamma(x,t)$ in the presence of a varying surface current $U(x,t)$ can be described using the following balance equation:

$$\frac{\partial \widetilde{\Gamma}}{\partial t} + \frac{\partial}{\partial x}\left[U(x,t)\widetilde{\Gamma}\right] = R \tag{1}$$

The right-hand side, $R$, of Equation (1) describes processes of surfactant diffusion, weathering, vertical mixing due to wave breaking and turbulence, etc., but here we neglect these processes. Assuming that $R = 0$, and $U(x,t) = U(x - Ct) + V$, a steady-state solution to Equation (1) can be written as follows:

$$\widetilde{\Gamma} = \frac{\Gamma_0}{1 - \frac{U(\xi)}{C-V}} \tag{2}$$

where $V$ is a uniform current on the water surface, $U(\xi) = U(x - Ct)$ denotes a non-uniform current, e.g., a surface component of the IW orbital velocity, propagating with a phase velocity $C$ in the positive direction of the x-axis, and $\Gamma_0$ is an undisturbed surfactant concentration, where $U(\xi) = 0$. If $V << C$, the concentration $\widetilde{\Gamma} \approx \Gamma_0\left[1 + \frac{U(\xi)}{C-V}\right]$, so that the concentration achieves its maximum values over the maxima of $U(\xi)$ (see Figure 1), which is over the troughs of pycnocline elevations for the IW basic mode. If $V$ is close to $C$ (a resonance condition), the concentration is small everywhere except for the vicinity of convergent points $\xi*$, where $U(\xi*) = |C - V|$ and $dU(\xi*)/dx < 0$). A steady-state solution to Equation (1) at resonance does not exist, and $\widetilde{\Gamma}$ grows constantly with time at the convergent points and can be limited to take into account relaxation/diffusion processes, for example. In the case of a resonance between a moving film and a sinusoidal IW $U(x - Ct) = U_0\sin(\kappa\xi)$, where $U_0$ and $\kappa$ are the IW amplitude and wave number, respectively, the surfactants will be strongly accumulated over IW rear slopes.

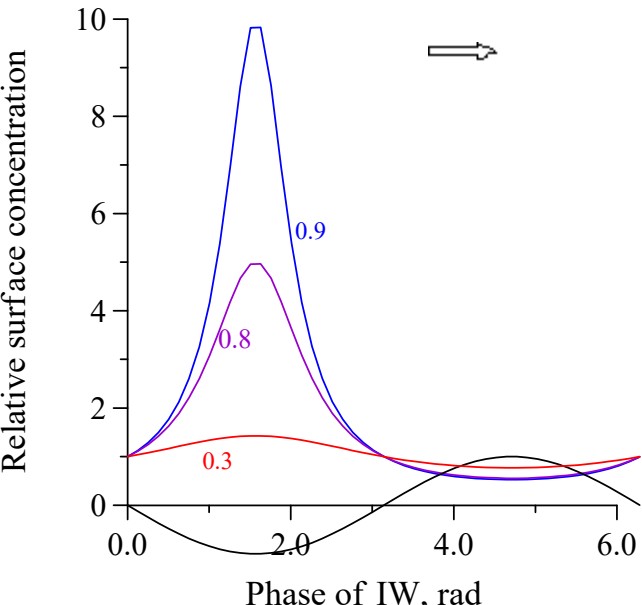

**Figure 1.** Surfactant concentration over a sinusoidal IW profile (black curve). Numbers near curves denote the values of $U_0/(C - V)$. The arrow indicates the IW propagation direction.

### 2.2. A Hydrodynamic Model of the Wind Wave Spectrum and Its Modulation

Wind waves can be described by the kinetic equation for the spectral density of the wave action $\widetilde{N}(k) \equiv N(k, x, t)$ of GCW in the following form (see, e.g., [6]):

$$\frac{\partial \widetilde{N}(k)}{\partial t} + [U(x,t) + c_{gr}(k)]\frac{\partial \widetilde{N}(k)}{\partial x} - k\frac{\partial U(x,t)}{\partial x}\frac{\partial \widetilde{N}(k)}{\partial k} =$$
$$= Q(\widetilde{N}(k), u*, \gamma(k, E(\widetilde{\Gamma}), \sigma(\widetilde{\Gamma})) \tag{3}$$

The wave action spectrum is related to the spectrum of wind wave heights as $\widetilde{F}(k) = \widetilde{N}(k) \cdot k / \omega(k)$. In Equation (3), $k$, $c_{gr}(k) = \partial\omega(k)/\partial k$ and $\omega(k) = (gk + \sigma k^3)$ are the wave number, group velocity and frequency of GCW, respectively; $g$ is the acceleration of gravity, $\sigma(\Gamma)$ is the surface tension. $Q \equiv Q(\widetilde{N}(k), u*, \gamma(k, E(\widetilde{\Gamma}), \sigma(\widetilde{\Gamma}))$ describes wind wave energy sources, sinks and nonlinear wave–wave interactions, $u*$ is the wind friction velocity, $\widetilde{\gamma}(k) \equiv \gamma(k, E(\widetilde{\Gamma}), \sigma(\widetilde{\Gamma}))$ is the wave damping coefficient [27], which, in turn, depends on the film elasticity $E(\widetilde{\Gamma})$ and surface tension and, in general, on some other surfactant film characteristics, e.g., diffusion coefficient, viscosity, etc. Hereafter, an upper tilde denotes the functions varying due to non-uniform currents; those with a subscript "0" correspond to undisturbed, steady-state values.

The r.h.s. of Equation (3) can be written as:

$$Q = \widetilde{\beta}^{eff}(k)\widetilde{N}(k) + P_a + I_{nl}\left[\widetilde{N}(k)\right] \tag{4}$$

In Equation (4), $\widetilde{\beta}^{eff}(k) = \beta(k, u*) - 2\gamma(k, E(\widetilde{\Gamma}), \sigma(\widetilde{\Gamma}))$, $\beta(k, u*)$ is the wind growth rate due to the Miles wave generation mechanism—the growth rate dependence on wind velocity and wind wave number can be found, e.g., in [28]—$P_a$ describes wind wave excitation due to atmospheric pressure pulsations—the so-called Phillips mechanism. Notably, that in further analysis, we neglect the variations of the wind growth rate over IW.

One of the most difficult problems is to describe the nonlinear terms $I_{nl}(\widetilde{N}(k))$ in the kinetic equation. The nonlinear term includes weakly nonlinear wave–wave interactions, and strongly nonlinear processes, such as wave breaking, responsible among others for the wind wave spectrum limitation. Wave–wave interactions are described by a collision integral which is extremely difficult to analyze. Wave breaking processes are also studied insufficiently and no analytical approach is developed to their description. Therefore, these complicated nonlinear processes are approximated by some empirical, usually algebraic expressions, similarly to [29], namely

$$I_{nl}\left[\widetilde{N}(k)\right] = -\alpha(k)\widetilde{N}^{n+1}(k) \tag{5}$$

Here, $\alpha(k)$ and $n$ are some empirical values. In the case when $\widetilde{\beta}^{eff}(k) < 0$, wind waves are not generated due to the Miles instability mechanism; thus, the Phillips excitation can generate wind waves, but of very small amplitudes. Therefore, neglecting the Phillips mechanism, one can consider that no waves are excited, in particular, with mm-scale wavelengths. Nevertheless, mm-scale waves appear, even $\widetilde{\beta}^{eff}(k) < 0$, because they can be generated by longer, cm–dm-scale waves if the Miles excitation condition is fulfilled for the latter. Thus, generated waves are known as bound waves, in particular, as parasitic capillary ripples (PCR) [15–20,26,30]. A simplified relation between the PCR wave number $k$ and the basic cm–dm-scale GCW $k_g$ can be written as follows:

$$k_g \approx \frac{g}{\sigma k} \tag{6}$$

In fact, the situation is more complicated, and the PCR wavelengths vary along the GCW profile [30], although here we ignore this fact and will suppose that PCR with a given wavelength is excited by GCW in a narrow band in the spectrum of GCW, centered around the wave number $k_g$. Then, we insert into the kinetic equation some function describing

PCR intensity as a function of the spectrum intensity of GCW with the wave number given by Equation (6).

Despite PCR being bound to the cm–dm-scale GCW we, nevertheless, described them as if they were free ripples when using the kinetic equation for the wind wave spectrum action. Therefore, both free wind-generated capillary ripples and parasitic ripples generated by longer steep GCW were indistinguishable in our approach. However, because we focused mostly on the analysis of wind wave spectrum modulation in the presence of surface films, free capillary ripples were not practically excited (for them, usually $\widetilde{\beta}^{eff}(k) < 0$) and all capillary waves were considered further as PCR.

For parasitic waves, it follows from a number of wave tank experiments that their parameters are characterized by very fast growth when the slopes of GCW are large enough. We assume this dependence in the power form as

$$k^2 \widetilde{F}(k) \propto \left[ k_g^2 \widetilde{F}(k_g) \right]^m \tag{7}$$

Hence, the source of PCR $\widetilde{I}(k_g)$ associated with GCW reads

$$\widetilde{I}(k_g) = \alpha(k_g) \widetilde{N}^m(k_g) \tag{8}$$

Then, the r.h.s. of Equation (3) for capillary waves when neglecting the pressure pulsations compared to $\widetilde{I}(k_g)$ looks like

$$Q(\widetilde{N}(k), u*, \gamma(\widetilde{\Gamma})) = \widetilde{\beta}^{eff}(k) \widetilde{N}(k) - \alpha(k) \widetilde{N}^{n+1}(k) + \widetilde{I}(k_g) \tag{9}$$

Suppression of capillary waves due to films can be characterized by the hydrodynamic contrast $K^{hydro}(k) = \frac{F_0^{nsl}(k)}{F_0^{sl}(k)}$. For mm-scale waves at low winds (of about 5 m/s and less), $\beta_0^{eff}(k) < 0$. In this case, when neglecting the second term in the r.h.s. of Equation (10), one can write the contrast $K^{hydro}(k)$ for mm-scale ripples as follows:

$$K^{hydro}(k) = \frac{N_0(k_g)|_{nsl}}{N_0(k_g)|_{sl}} = \frac{I_0(k_g)|_{nsl}}{I_0(k_g)|_{sl}} \frac{\beta_0^{eff}(k)|_{sl}}{\beta_0^{eff}(k)|_{nsl}} = \left[ K^{hydro}(k_g) \right]^m \frac{\beta_0^{eff}(k)|_{sl}}{\beta_0^{eff}(k)|_{nsl}} \tag{10}$$

where $K^{hydro}(k_g)$ denotes the contrast for basic cm–dm-scale GCW generating PCR, and $nsl, sl$ mean non-slick (without film) and slick (with film) areas, respectively.

Variations of the spectrum of mm-scale ripples under the action of IW contain several components. They are kinematic (straining) variations due to IW orbital velocities, variations of the damping coefficient due to the redistribution of surfactant concentration as well as variations of the wind wave growth (we neglect this effect in the following as insignificant), and variations of the PCR source due to kinematic modulation of cm–dm-scale waves. The hydrodynamic modulation transfer function is $m_h = m_h(k, \kappa, C, \ldots)$, which characterizes the wind wave spectrum variations $m^{hydro} \frac{U_0}{C} e^{i\kappa(x - Ct)} = \frac{F(k) - F_0(k)}{F_0(k)} = \frac{\Delta F(k)}{F_0(k)} e^{i\kappa(x - Ct)} = \frac{\widetilde{N}(k) - N_0(k)}{N_0(k)} = \frac{\Delta N(k)}{N_0(k)} e^{i\kappa(x - Ct)}$ The definition of $m^{hydro}$ usually implies that $\Delta F(k) << F_0(k)$ and can be written for capillary waves when using Equations (3)–(9), as follows

$$m^{hydro}(k) = \left[ -\frac{k}{N_0(k)} \frac{\partial N_0(k)}{\partial k} - i \frac{\Delta \beta^{eff}(k)}{\Omega} \frac{C}{U_0} - i \frac{\partial I_0(k_g)}{\partial N_0(k_g)} \Delta N(k_g) \cdot \frac{C}{U_0 \Omega} \right] \cdot$$
$$\left[ 1 - \frac{c_{gr}(k)}{C} + i \frac{\beta_0^{eff}(k) - (n+1)\alpha(k)N_0^n(k)}{\Omega} \right]^{-1} \tag{11}$$

Here, $\kappa$, $C$ and $\Omega$ denote the wavenumber, phase velocity and frequency of IW, respectively, $\Delta\beta^{eff}$ and $\Delta N$ are the amplitude of variations of $\beta^{eff}$ and spectral wave action due to IW, respectively.

The first two terms in the first bracket of the r.h.s. of Equation (11) are due to the straining of free wind ripples and due to variations of the wave damping coefficient associated with surfactant concentration modulation, respectively; the third term corresponds to modulation of the source of parasitic ripples. The term $(n+1)\alpha(k)N_0^n(k)$ in Equation (11) can be neglected at low wind ripples intensity, as discussed above.

The hydrodynamic modulation transfer function for GCW, i.e., at a wave number $k_g$ looks like Equation (11), except for the term describing the source due to longer wave components, namely

$$m^{hydro}(k_g) = \left[ -\frac{k_g}{N_0(k_g)}\frac{\partial N_0(k_g)}{\partial k} - i\frac{\Delta\beta^{eff}(k_g)}{\Omega}\frac{C}{U_0} \right] \cdot$$
$$\left[ 1 - \frac{c_{gr}(k_g)}{C} + i\frac{\beta_0^{\text{eff}}(k_g) - m\alpha(k_g)N_0^{m-1}(k_g)}{\Omega} \right]^{-1} \tag{12}$$

### 2.3. A Model of Microwave Radar Backscatter from the Sea Surface

Total normalized radar cross-section (NRCS) in the Ka-band from a wavy water surface can be written as a combination of polarized (Bragg) and non-polarized (non-Bragg) components, BC and NP, respectively (see, e.g., [31,32]):

$$\sigma^{pp} = \sigma_{BC}^{pp} + \sigma_{NP} \tag{13}$$

The term, $\sigma_{BC}^{pp}$ is determined by the spectrum $\widetilde{F}(k_{\text{br}})$ of mm-scale wind ripples, including both free waves and parasitic capillary ripples, at the Bragg wavenumber [33] $k_{br} = 2k_e\sin\theta$ ($k_e$ and $\theta$ are the wavenumber and the angle of incidence of microwaves), and by the scattering coefficient $g_{pp}^2(\theta)$, depending on the polarization (p) of the emitted/received signal, the incidence angle, and dielectric properties of water (see, [33]), and can be written as

$$\sigma_{BC}^{pp} = 16\pi k_e^4 g_{pp}^2(\theta)\,\widetilde{F}(k_{br}) \tag{14}$$

The second, non-polarized component in Equation (13), $\sigma_{NP}$, is determined by scattering from quasi-specular facets on the wind wave profile and is obviously equal for vertical and horizontal emitted/received polarizations, VV and HH, respectively. The appearance of these facets is related to high steepness/curvature parts of the nonlinear structures, such as bulge/toes and partly to parasitic capillary ripples on the profile of steep cm-dm-scale wind waves (see, e.g., [18–20,26]).

A relationship between NP radar return and characteristics of the nonlinear structures has not been established thus far. However, some results of our wave tank experiments, which will be published elsewhere, indicate that the NP Ka-band radar return can be described by a power function of the steepness of basic cm–dm-scale GCW. Accordingly, we suppose that

$$\sigma_{NP} \sim \left[ k_g^2\widetilde{F}(k_g) \right]^q \tag{15}$$

where $q \approx 5$.

We define the radar slick contrasts as a ratio of total radar return, BC and NP in non-slick (*nsl*) and slick (*sl*) areas, according to:

$$K^{pp} = \frac{\sigma^{pp}|_{nsl}}{\sigma^{pp}|_{sl}}, \quad K_{BC}^{pp} = \frac{\sigma_{BC}^{pp}\big|_{nsl}}{\sigma_{BC}^{pp}\big|_{sl}} = \frac{F_0(k)|_{nsl}}{F_0(k)|_{sl}}, \quad K_{NP} = \frac{\sigma_{NP}|_{nsl}}{\sigma_{NP}|_{sl}}, \tag{16}$$

It can be shown clearly that the slick contrast for the total radar return is related to the partial, NP and BC contrasts according to the formula:

$$K^{pp} = \chi_{BC}(\Gamma_0) \frac{\sigma_{BC}^{pp}\big|_{nsl}}{\sigma_{BC}^{pp}\big|_{sl}} + \chi_{NP}(\Gamma_0) \frac{\sigma_{NP}\big|_{nsl}}{\sigma_{NP}\big|_{sl}}, \tag{17}$$

where $\chi_{BC}(\Gamma_0) = \frac{\sigma_{BC}^{pp}\big|_{sl}}{\sigma^{pp}\big|_{sl}}$ and $\chi_{NP}(\Gamma_0) = \frac{\sigma_{NP}\big|_{sl}}{\sigma^{pp}\big|_{sl}}$ are the relative Bragg and non-polarized radar backscatter components in total radar return at a given mean surfactant concentration $\Gamma_0$.

Variations of radar return due to internal waves $\Delta\sigma^{pp}$ can be characterized by the radar modulation transfer function (RMTF) $m^{radar\_pp}$, according to its definition, namely

$$\Delta\sigma^{pp}/\sigma^{pp} = m^{radar\_pp} \frac{U_0}{C} e^{i\kappa(x - Ct)}, \tag{18}$$

RMTF can be described as follows:

$$m^{radar\_pp} = \chi_{BC}(\Gamma_0) \cdot m^{hydro}(k) + \chi_{NP}(\Gamma_0) \cdot qm^{hydro}(k_g) \tag{19}$$

## 3. Wave Tank and Field Experiments

### 3.1. Wave Tank Experiments

The wave tank experiments were carried out in the Oval Wind Wave Tank of the Institute of Applied Physics, RAS. A sketch of the experiments is shown in Figure 2. The tank was half-filled with freshwater (density of 0.997 g/cm$^3$); then, saltwater was slowly poured through special holes in the tank bottom. The thickness of each layer was about 14 cm. IWs were generated by a vertical paddle oscillating around a horizontal axis located at the interface between the salt- and freshwater layers. IWs were absorbed at the end of the working section of the tank using a special absorbing system consisting of three parts. First, two long (1.5 m length) false vertical walls were installed at a small angle to the tank walls in order to smoothly reduce the tank width up to about 2/3 of its initial value; then, there was a system of submerged beaches and, finally, a system of plastic sheets (about 10 sheets) suspended vertically across the tank at distances of about 10 cm between the sheets. The reflection coefficient for the IWs was less than 10%. IW amplitudes were recorded using resistance wire gauges mounted vertically beneath the water surface and centered at the inter, and the amplitudes were varied in the range of 3–5 mm. Surface waves were recorded in the middle tank plane with resistance gauges similar to IW gauges, but adapted for freshwater. Wind waves were generated with a ventilator (1). Wind speeds were measured in the center of the wind channel cross-section using a small propeller sensor; the experiments were carried out at mean wind speeds of 2 and 2.5 m/s.

The wind drift velocity was measured using small floats (less than 2 mm thick) from a sponge with neutral buoyancy drifting in the water surface layer. The measured average wind drift velocities at wind speeds of 2 m/s and 2.5 m/s were 5–6 cm/s and 6–7 cm/s, respectively.

Two cases were studied in experiments at different density jumps between the upper and lower water layers. The first case corresponded to "fast" IWs, propagating with phase velocities essentially exceeding the wind drift velocities ("non-resonance case"). The second set of experiments was realized with "slow" IWs, for which phase velocities were close to the wind drift velocities and hence a resonance modulation of surfactants occurred (see Section 4).

Oleic acid (OLE) was used to produce a monomolecular film on the water surface. Elastic properties of the surfactant were close to those for biogenic films on the sea surface and for thin films of crude oil [34]. OLE was dissolved in ethanol (the solution volume concentration was 10 mg/mL) to improve spreading and was applied on the water through a dropper; its nozzle was positioned at a height of 2–3 mm over the water surface. The

dropping period was varied in the range from 2 s to 10 s. The dropper was located at a distance of 10 m from the working area (see Figure 2) and low variations of Ka-band radar return in the absence of IW indicated that OLE formed a uniform film in the study area. Surfactants were continuously skimmed by paper napkins at the end of the tank where the absorbing system was located.

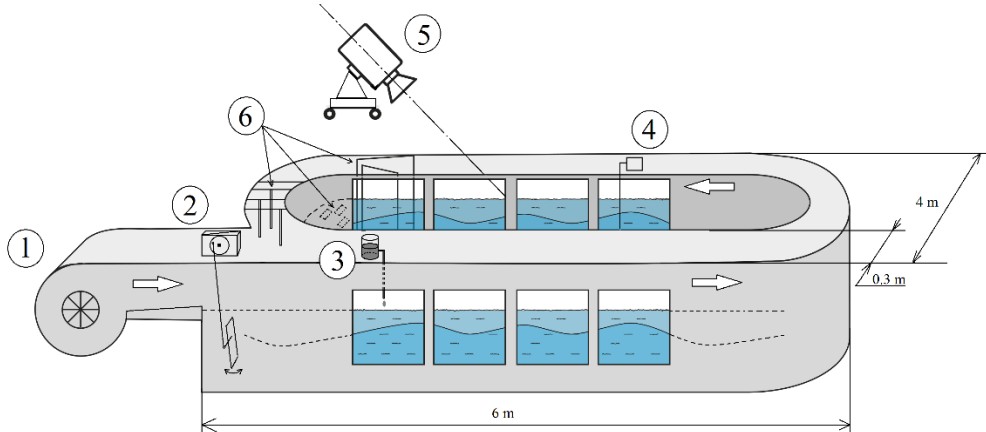

**Figure 2.** The Oval Wind Wave Tank of IAP RAS (1—a ventilator for wind flow generation; 2—an IW maker; 3—a dropper; 4—wire gauges of IWs and surface waves; 5—Ka-band radar; 6—a wave absorbing system).

Film samples were collected from the water surface using nylon nets. The damping coefficient and the wavelength of capillary standing waves (Faraday ripples) were measured for the sampled films with a method of parametric waves (see [35]) and the surface tension coefficient (STC) and the film elasticity were retrieved. Then, using previously obtained dependencies of these parameters on OLE concentration, the latter was retrieved. These dependences for the film elasticity and film pressure (the difference between STC values for clean and film covered surfaces) are shown in Figure 3 (see also [32]). The water surface in the tank before the experiments was cleaned with paper napkins until the STC for film samples differed from the tabulated values for clean water by at least 3% or less. To study variations of surfactant concentration over an IW profile, net samples were collected simultaneously over different phases of IW, namely, over IW crests, troughs and slopes.

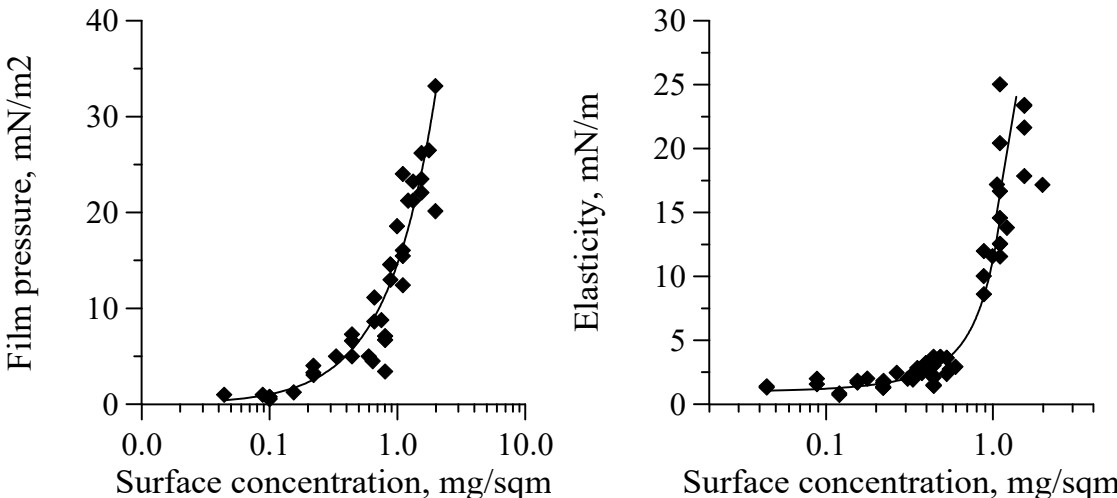

**Figure 3.** Film pressure (**left**) and elasticity (**right**) vs. surface concentration of OLE. Measurement by the method of parametric waves at a frequency of 25 Hz. Solid curves—least squares interpolations.

Variations of surfactant concentration over IW can be retrieved when estimating STC and elasticity values from investigations of film samples using the parametric method. It is also important to know a mean surfactant concentration which can be determined when analyzing net samples taken without IW. To make this determination more reliable, we used additional considerations to estimate $\Gamma$ values. We assumed that a steady film was formed in the working area as a result of balance between the OLE inflow due to dropping and the outflow due to wind drift. Therefore, we estimated the mean surfactant concentration as $\Gamma \sim C_s V_d / (T_d V_{dr} L_T)$, where $V_d$ is the volume of droplets, $C_s$ is the volume concentration of surfactants in the alcohol solution, $V_{dr}$ is the drift velocity, $T_d$ is the period of droplets, and $L_T$ is the tank width. Thus, estimated concentration values were found to be consistent with measurements.

The Ka-band radar used in our experiments was a scatterometer operating in a continuous wave regime. The microwave radar frequency was about 35 GHz and was modulated within the band of 100 MHz. A radar is a coherent system measuring the amplitude and phase of scattered microwaves that allows one to investigate the Doppler spectrum of radar return. The radar antenna beam width was about $4°$. Although the scatterometer is a single polarized device, it can be utilized for measurements at vertical or horizontal co-polarizations of transmitted/received microwaves, VV or HH polarizations, respectively. VV and HH polarization measurements can be arranged consecutively, by simply turning the instrument around the antenna axis by 90 degrees. Notably, the radar antenna patterns in two orthogonal planes are slightly different, and this leads to small (about 10%) differences between the radar backscatter at VV and HH polarizations; however, this was insignificant in our experiments. The radar was mounted at a height of about 1 m over the tank looking at an incidence angle $\theta = 50°$ in the upwind direction. The cross-tank and along-tank sizes of the radar footprint on the water surface were about 17 cm and 30 cm, respectively. This allowed us to analyze ripple variability along the IW profile.

In order to reduce parasitic reflections from metal constructions in the laboratory, the top of the tank below the radar was covered with absorbing sheets, and a gap of about 15 cm in width was left between the sheets to limit the radar beam width. An example of the Doppler radar spectrum is shown in Figure 4. The Doppler spectrum could be divided into two parts. One part was characterized by a peak with a frequency offset at about 50–60 Hz, which corresponded to scatterers moving with velocities close to the phase velocities of free gravity-capillary waves with the Bragg wavelength. The low-frequency part was due to side lobes of the radar antenna and due to specular reflections from the walls and metal constructions in laboratory. Notably, the intensity of the low-frequency peak did not depend on the presence of films on the water surface in practice, whereas the Bragg peak could be strongly suppressed in the presence of films. In order to reduce the influence of the low-frequency part of the radar return, we eliminated this part, approximating it in the range of Bragg peak by a smooth curve, as shown in Figure 4 (dashed curves). Then, this part was deducted from the total Doppler spectrum in order to extract the spectrum in the Bragg frequency range. Obviously, there was some arbitrariness in choosing the approximating spectrum, but the final results after testing different approximations were found not to be strongly dependent on specific approximating curves.

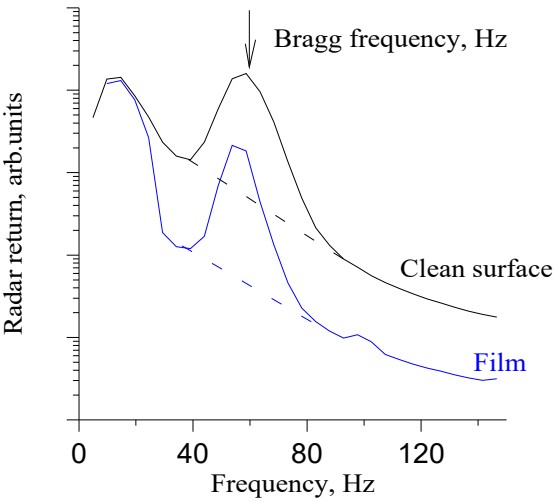

**Figure 4.** The Doppler spectra of the radar return.

### 3.2. Field Experiments

The observations were carried out from an oceanographic platform located 500 m from the shore in a coastal zone of the Black Sea (see Figure 5). The water depth $H$ near the platform was 30 m.

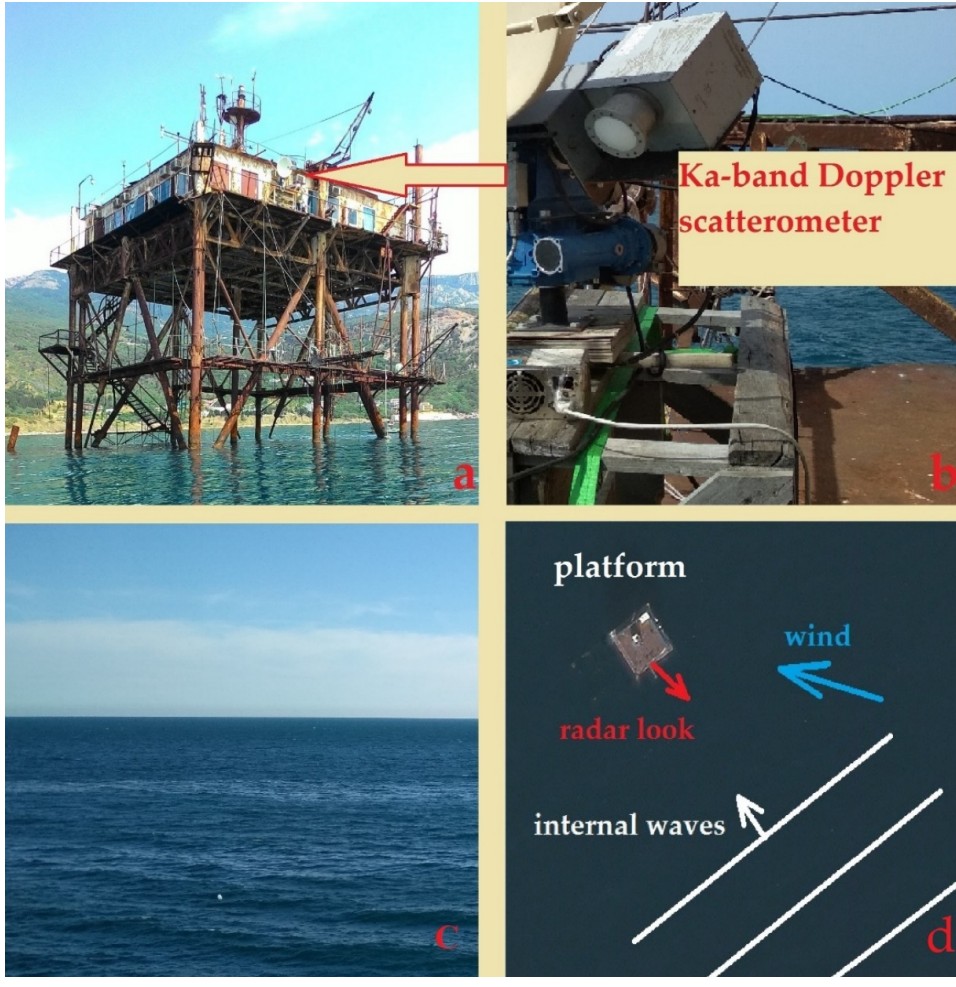

**Figure 5.** The oceanographic platform on the Black Sea (**a**); Ka-band scatterometer (**b**); banded slicks due to IWs (**c**); and a sketch of radar observations (**d**).

Radar observations were performed using the same Ka-band scatterometer as was used in the wave tank experiments. The scatterometer was mounted at the upper deck of the platform at a height of about 12 m looking in the south-east direction at an incidence angle of 60°, as shown schematically in Figure 6. Some auxiliary observations of the sea surface were also performed with an X-band navigation marine radar, MRS-1000. IWs were recorded using an ADCP "Workhorse Monitor" 1200 kHz mounted on the platform at a depth of 0.5 m in the upper water layer and looking downward. Wind velocity/direction were measured with an ultrasound anemometer.

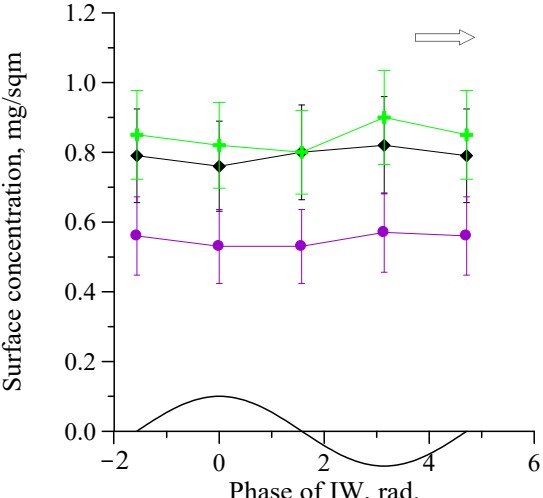

**Figure 6.** Distributions of surfactant concentration over IW profile in the first experimental set; the black curve at the bottom of the figure is the IW profile. The arrow indicates the IW propagation direction, and the curves/dots of different colors correspond to different average surfactant concentrations: 0.8 mg/m² (black and green), and 0.6 mg/m² (violet). Period, wavelength, and amplitude are 9 s, 200 cm, and 5 mm, respectively.

### 3.3. Data Processing

The radar modulation transfer function $m^{radar\_VV}$, which we denote further for brevity as $m(\Omega)$, was analyzed in the experiments as follows (see [36]):

$$m(\Omega) = -\frac{C}{\overline{P}} \cdot \frac{\overline{P(\Omega)U^*(\Omega)}}{|U(\Omega)|^2} = \frac{1}{\overline{P}} \frac{\overline{P(\Omega)Z^*(\Omega)}}{|Z(\Omega)|^2} \cdot \frac{sh(\kappa h)}{\kappa} = |m| \cdot e^{i\varphi_m} \qquad (20)$$

In Equation (20), $P(\Omega)$, $U(\Omega)$ and $|Z(\Omega)|$ denote Fourier components of the received radar signal power, IW orbital velocity, and pycnocline elevation due to IWs, respectively; the horizontal bar denotes averaging over the IW period. $|m|$ is the RMTF magnitude, and $\varphi_m$ is the RMTF phase, which is the difference between the phase of radar backscatter variations and the IW pycnocline elevations. Note that $\varphi_m = 0$ if the radar return maxima are located over the minima of the IW orbital surface velocity, or equivalently, over the crests of the pycnocline elevations. The duration of time records in the experiment were about 10 min, and the sampling rate was 1250 Hz. The radar intensity over a time interval of 1–2 s was calculated when integrating the radar Doppler spectrum in the frequency band of 50–150 Hz. The obtained time series of the radar intensity and pycnocline displacement oscillations were divided into 3 min realizations; for each realization, the Fourier spectra were calculated, and then RMTF was calculated according to Equation (20).

To estimate the errors of MTF values, the coherence $\Upsilon^2(\Omega)$ was calculated as (see [36]):

$$\gamma^2(\Omega) = \frac{\overline{|P(\Omega)U^*(\Omega)|}^2}{|P(\Omega)|^2|U(\Omega)|^2}, \qquad (21)$$

The coherence characterizes a ratio between the variance of the radar return fluctuations, correlated with IW and the total variance of the radar return fluctuations [25]. The error of MTF increases with a decrease in the coherence, and estimates of the magnitude and phase of the MPF are unreliable if the coherence is small.

## 4. Results

### *4.1. Wave Tank Experiments*

#### 4.1.1. Modulation of Surfactant Concentration Due to IW

The first experimental set was with the density difference between the lower and upper layers of 0.07 g/cm$^3$, and the estimated IW phase velocity was 23 cm/s. Amplitudes of IWs were about 5 mm. Figure 6 illustrates OLE concentrations over IW crests, troughs, forward and rear slopes. The relative variations in surface concentration related to the mean OLE concentrations were small, about 5% to 10%, which is comparable or less than the measurement error. However, analyzing several concentration profiles, one could conclude that the maximum surfactant concentration was located over IW troughs.

In the second experimental set, the density difference between the lower salt layer and the fresh upper layer was very small, about 0.005 g/cm$^3$; therefore, the phase velocity of long IWs was comparable with the wind drift velocity on the water surface, i.e., the IW phase velocity was 7 cm/s, for an IW length of 175 cm and IW period of 25 s. This IW phase velocity is smaller than the minimum group velocity of GCW; therefore, no resonance condition between IW and GCW is fulfilled. However, we characterized this case as "quasi-resonance" in terms of a resonance between IW and a film moving due to wind drift. Figure 7 demonstrates the measured surfactant concentration variations over IW profiles at different dropping rates. At a wind speed of 2 m/s, surfactant concentration maxima were observed mostly over IW troughs (phase $\pi$); at a wind speed of 2.5 m/s, the maximum surface concentration was observed mostly over the rear slopes of IW. Sampling only in four points on the IW profile did not allow us to study the process of the maximum concentration shift from the trough to the rear slope with better resolution in the IW phase. We should emphasize that, in one case out of five, the concentration maximum was over IW troughs at wind speeds of 2.5 m/s, and over rear IW slopes at wind speeds of 2 m/s, although we did register strong variations of OLE concentration which exceeded the error of measurement. The relative variations of surface concentration normalized by their means are presented in Figure 8 for two case studies.

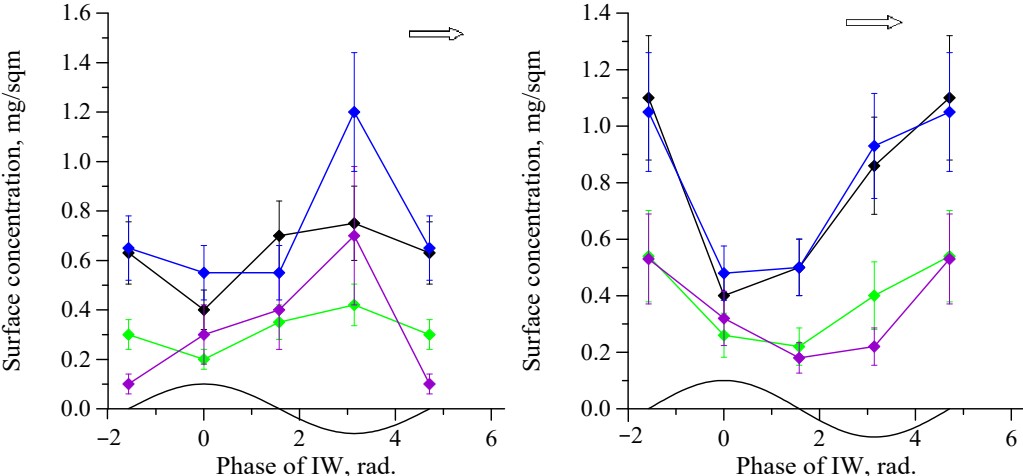

**Figure 7.** Same as in Figure 6, but for quasi-resonance cases. Curves/dots correspond to different average surfactant concentrations and IW amplitudes. Left panel: black—0.6 mg/m$^2$, IW amplitude of 3 mm; blue—0.8 mg/m$^2$, and 5 mm; violet—0.4 mg/m$^2$, and 5 mm; green—0.3 mg/m$^2$, and 3 mm; Right panel: black and blue—0.8 mg/m$^2$, IW amplitude of 5 mm; violet and green—0.3 mg/m$^2$, and amplitude of IW—3 mm. Period and wavelength are 25 s and 175 cm, respectively.

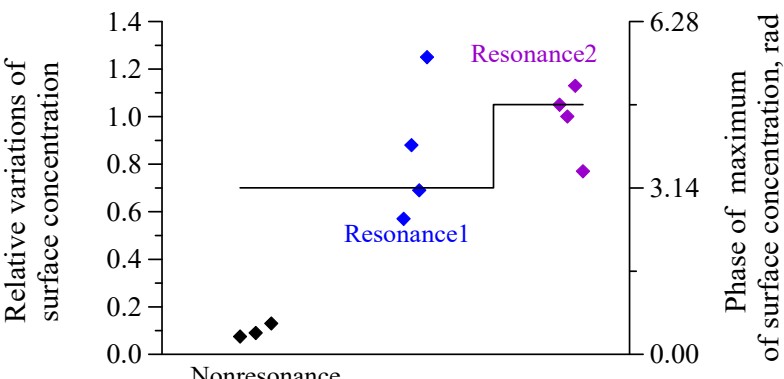

**Figure 8.** The relative variations of surface concentration at the resonance and the non-resonance. Black solid line—phase of maximum surface concentration.

4.1.2. Ka-Band Radar Probing of Wind Waves

Wind Wave Spectra

Typical spectra of wind waves at low wind velocities of 2 m/s and 2.5 m/s are shown in Figure 9.

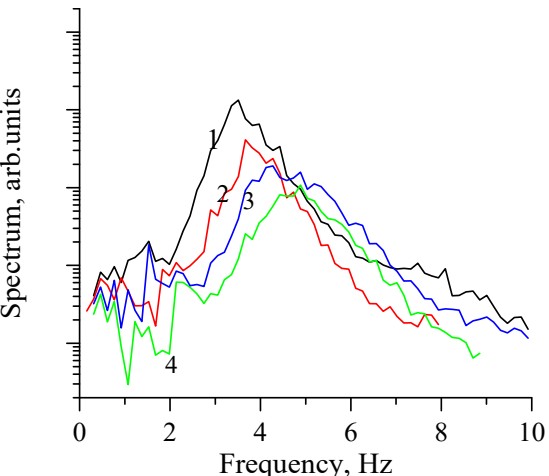

**Figure 9.** Wave height spectra (1—clean water, 2.5 m/s; 2—OLE concentration 0.6 mg/m$^2$, 2.5 m/s; 3—clean water, 2 m/s; 4—OLE concentration 0.6 mg/m$^2$, 2 m/s).

The spectra in Figure 9 indicate that the spectrum peaks corresponding to the wavelengths varied from about 7 cm to 25 cm. It is known that GCW within this range are characterized by the appearance of nonlinear structures such as bulges/toes and parasitic capillary ripples at GCW crests and forward slopes, respectively. Therefore, we expect that these structures will significantly contribute to Ka-band radar backscatter. Figure 9 also shows that dominant wind waves of cm–dm-scales are suppressed in the presence of surfactant films; the contrasts values are about 2, and this suppression can result in the damping of parasitic ripples accordingly.

Radar Contrasts

The radar contrasts as functions of OLE concentration at two wind speeds are presented in Figure 10 at the non-resonance and resonance between the IW and the film drift.

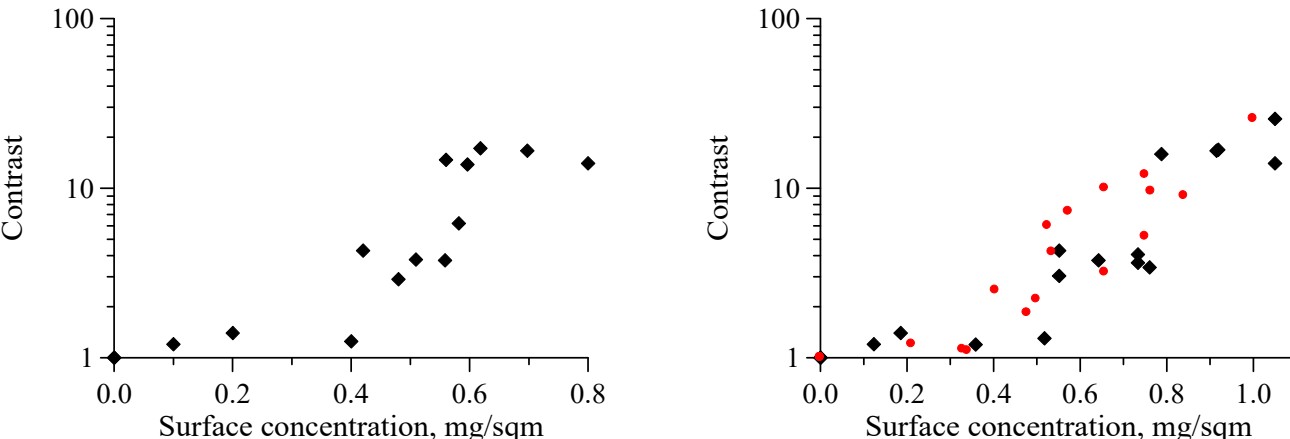

**Figure 10.** Contrast vs. surface concentration of OLE for a non-resonance (**left**) and resonance (**right**) case when the IW and film drift velocities are close to each other. Wind velocities are of 2.5 m/s (black) and 2 m/s (red).

From Figure 10, one can conclude that the mean contrast characterizing the suppression of radar backscatter increases with surfactant concentration, but is not practically affected by a relationship between the IW phase velocity and film drift. This could be expected, particularly for the case of not-too-strong variations of radar backscatter due to IW when the radar return modulation can be described as sinusoidal, which is removed after the radar backscatter averaging.

Radar Modulation Transfer Function

The magnitude and phase of RMTF as well as the coherence are shown in Figure 11 for two case studies: resonance and non-resonance modulation of surfactant concentration. Remarkably, the obtained dependencies for RMTF magnitude and phase are completely different. For the case when the IW phase velocity is significantly larger than the film drift velocity, the RMTF magnitude weakly depends on the presence of surfactants being practically constant up to 0.6 mg/m$^2$ and is slightly reduced at higher OLE concentrations. The RMTF phase is also practically constant, and is close to $-\pi/2$; therefore, the radar return maxima are located over the rear slopes of IW pycnocline elevations. The coherence values exceeded 0.5, which is a rather high correlation, so a confidence interval in our cases is less than 25%. On the contrary, the RMTF magnitude for "slow" IWs, which are in resonance with film drift velocity, grows with surfactant concentration being rather small at zero/low $\Gamma$ values, but exceeding the non-resonance RMTF magnitudes at sufficiently high surfactant concentration values. The RMTF phase for clean surfaces at the resonance is close to zero, i.e., the maximum of the radar return is over the IW crests, the phase of modulation increases with surface concentration, and the maximum radar return is shifted to the forward slope (2 m/s) or to the trough (2.5 m/s).

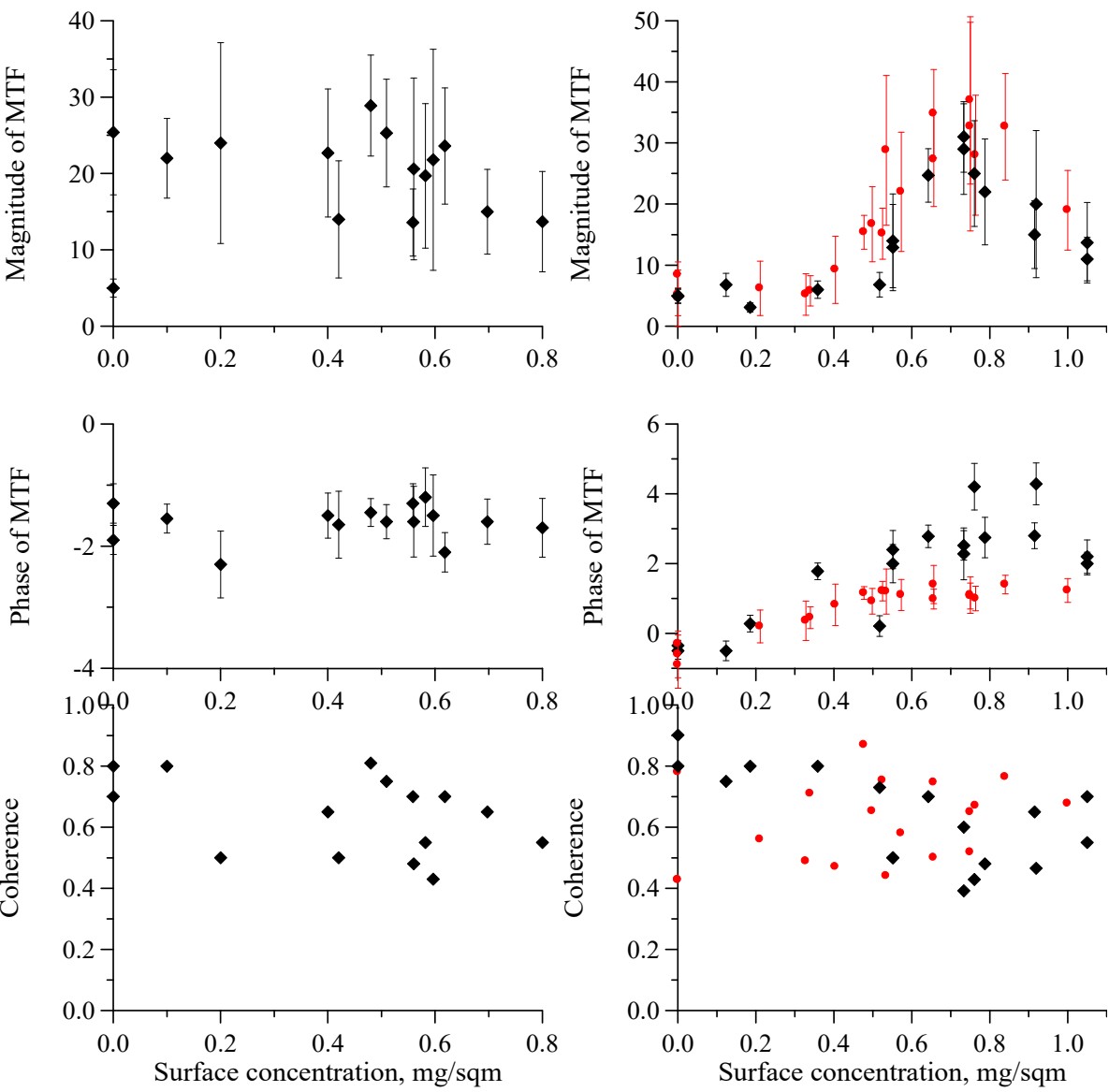

**Figure 11.** Radar MTF and coherence vs. surface concentration of OLE. Non-resonance (**left**) and resonance (**right**) cases: wind velocity 2.5 m/s (black), and 2 m/s (red). The 90% confidence intervals were calculated according to [36].

### 4.2. Field Observations

Typical IW signatures in the study area appeared at gentle wind conditions in the form of several quasi-parallel slick bands moving towards the shore. The slick bands were clearly seen by eye from a distance because of the suppression of small-scale wind waves, and near the platform because of enhanced water turbidity, debris in the surface water layers, and dust and pollen on the water surface, thus indicating the presence of biogenic surfactant films. The typical distances between the slick bands near the platform estimated from radar panoramas were about 100 m. The presence of surfactant films in slick bands was confirmed in our previous numerous measurements in the study area when analyzing variations of the surface tension values between slick and non-slick areas (see, e.g., [13]). In the reported observations, the wind velocity was about 4 m/s, and the subsurface mean current velocity measured at a depth of 1 m was about 17 cm/s. The thickness of the upper warm water layer was about 6–7 m, and the temperature difference between the upper and deeper water layers was about 10 °C. The wind speed and marine current directions, as well as Ka-band radar and the IW propagation directions, were roughly co-directional. An IW packet recorded with the ADCP due to oscillations of ultrasound scattering layers is

depicted in Figure 12; the estimated periods of IWs were about 5–6 min. A time series of radar backscatter in Figure 12 (the upper panel) illustrated strong radar return suppression in the slick bands. The dips of Ka-band radar backscatter in the slick bands were recorded over the rear slopes of the IW profile. The radar backscatter damping ratio, which is the ratio of non-slick-to-slick radar return intensities, varied in the range from about 10 to 100; this is typical for biogenic film slicks (see, e.g., [13]).

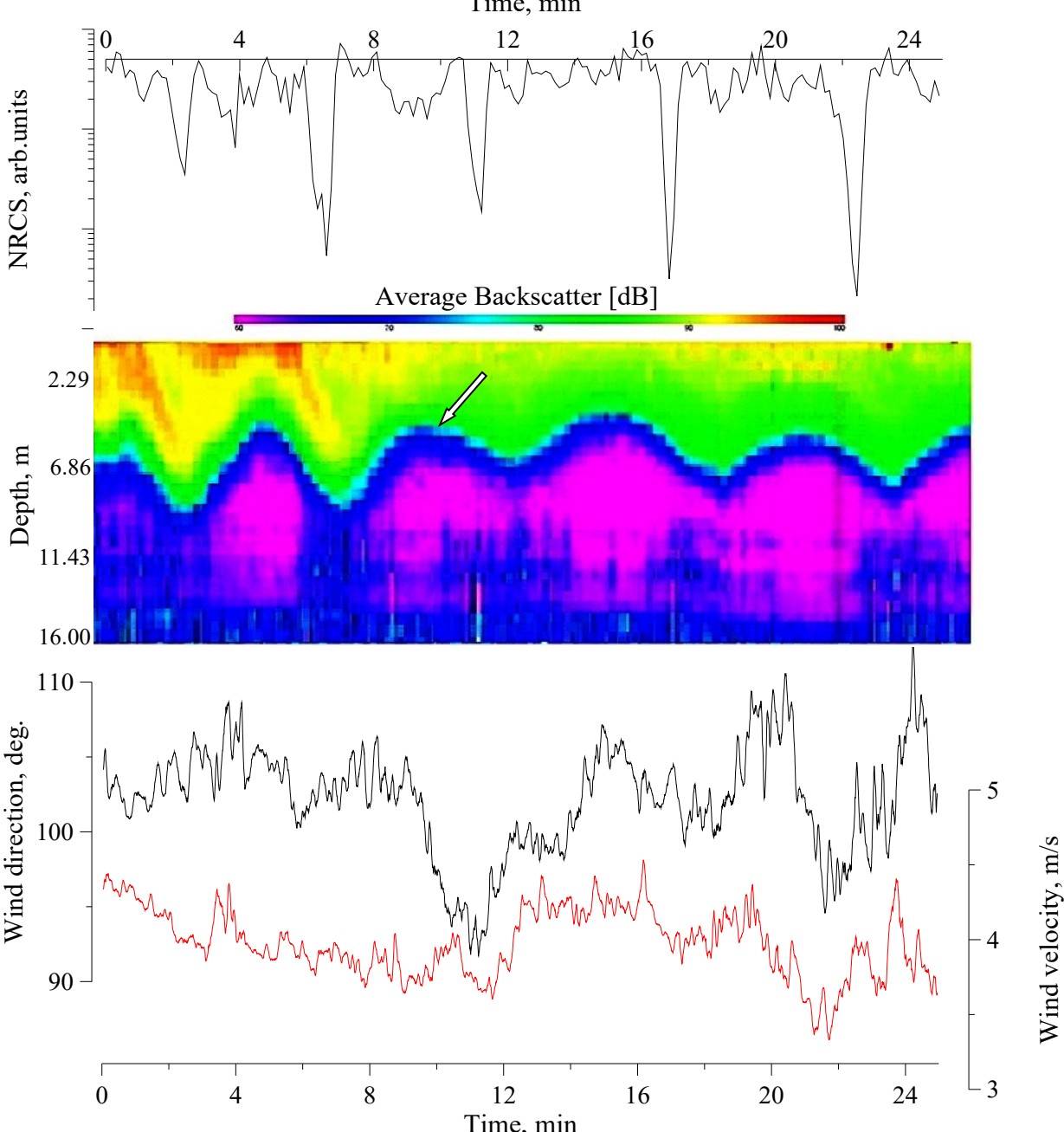

**Figure 12.** Internal wave profile from acoustic (ADCP) scattering data (**middle**), variability of Ka-band NRCS (**top**), wind velocity and direction (**bottom**): red curve—velocity, back curve—direction. The arrow indicates a rear slope of the IW.

## 5. Discussion

As mentioned above, the Ka-band of the radar backscatter contains two components—Bragg and non-polarized components. Bragg scattering can be determined both by free capillary waves directly excited by wind and by bound waves—parasitic capillary ripples

had wavenumbers which also fitted the Bragg resonance condition in the Ka-band range. The NP component was determined by the nonlinear structures such as bulges/toes, and possibly some initial part of PCR wave trains at a sufficiently high steepness of carrying cm–dm-scale GCW. For further interpretation of the experimental results, we discuss first how strongly Bragg scattering in our experiments contributed to the total radar return at VV polarization. To estimate this contribution, some auxiliary (without IW) investigations were performed to record the Ka-band radar backscatter at VV and HH polarizations; the HH-polarized measurements were carried out by simply turning the radar antenna around its axis by 90 degrees. Relationships between microwave scattering at two co-polarizations are usually characterized by the polarization ratio (PR), which is a ratio of VV- to HH-polarized radar returns. The polarization ratio as a function of surfactant concentration depicted in Figure 13 demonstrates that in our experiments, the PR values for a clean water surface are close to those predicted by Bragg theory for VV-polarized radar return. This means that Bragg scattering mostly determines Ka-band VV-polarized backscattering, whereas the role of the Bragg mechanism decreases when the surfactant concentration grows (see Figure 13, right panel, illustrating the ratios of Bragg and non-polarized components to total VV backscatter as functions of OLE concentration). One should note that the radar backscatter at high OLE concentrations tended to non-polarized scattering. This is consistent with the results of wave tank studies of the action of surface films on the profile of steep cm–dm-scale GCW [37,38]. The effect revealed in [37,38] is explained by the fact that PCR is more effectively suppressed by surface films than bulge/toe structures, and the latter determine mostly non-polarized scattering.

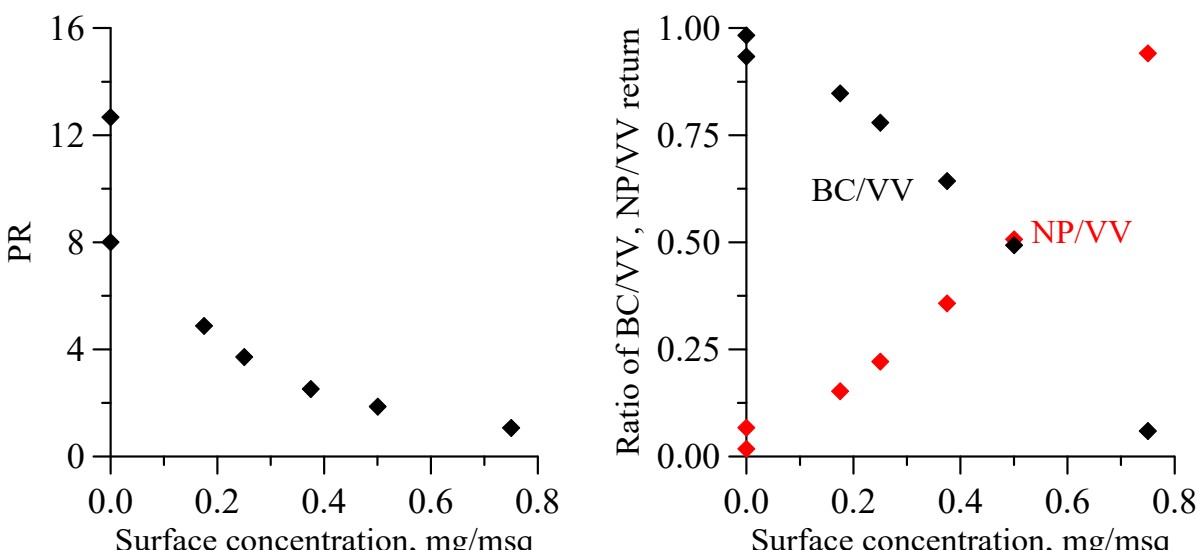

**Figure 13.** Polarization ratio vs. surfactant concentration at a wind velocity of 2 m/s (**left**) and relative Bragg and non-polarized components normalized by the total VV radar backscatter (**right**).

Let us now consider VV polarized radar contrasts as functions of OLE surface concentration. The total VV-polarized contrasts calculated according to the model described above using Equations (11), (14)–(17) are depicted in Figure 14. For our calculations, we used the following expressions for the wind growth rate [28]:

$$\beta(k, u*) = 0.04 k^2 u*^2 / \omega(k), \tag{22}$$

and for the wave damping coefficient [27]:

$$\gamma(k, E) = 2\nu k^2 (1 - X + XY)/(1 - 2X + 2X^2) \tag{23}$$

where $\nu$ is the kinematic water viscosity, $X = \frac{Ek_g^2}{\rho(2\nu)^{1/2}\omega^{3/2}}, Y = \frac{Ek_g}{4\rho\nu\omega}$, and we assumed that $m = 3$ and $n = 1$.

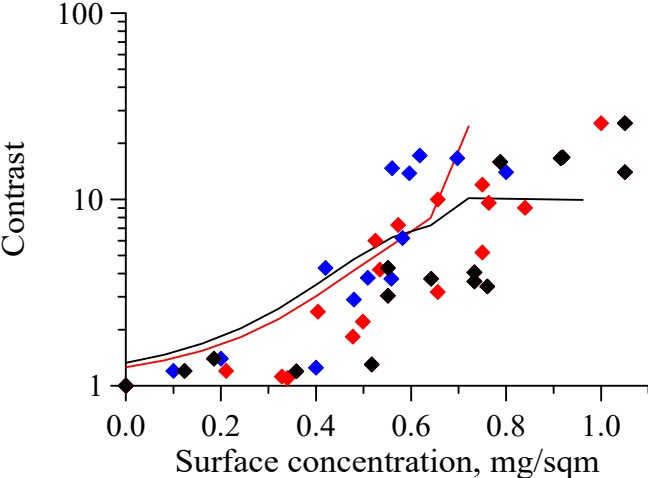

**Figure 14.** Total radar backscatter contrast. Black and red curves are model contrasts for wind velocities of 2.5 m/s, and 2 m/s, respectively: symbols—experiment.

The relative intensities of Bragg and non-polarized components in Equation (17) were taken from the experiment according to Figure 13. Note that radar return suppression is determined solely by the suppression of cm–dm-scale GCW contributing both to Bragg and NP scattering, because Bragg scattering from free mm-scale Bragg waves is absent. According to the theory, wind waves in the cm–dm wavelength range of the spectrum are suppressed by about 1.5–2 times, which is approximately consistent with the experiment (see, Section 4). One should emphasize that the obtained Ka-band radar contrast is significantly larger than the hydrodynamic suppression of cm–dm-scale GCW, and this occurs because of the high sensitivity of nonlinear structures on GCW profiles to small reductions in the spectrum intensity of these GCW (a "cascade suppression mechanism").

One can conclude from Figure 14 that the phenomenological model described above is consistent with the experiment.

The results of applying the model of radar backscatter modulation due to IW are presented in Figure 15. Here, the RMTF magnitude and phase as functions of surfactant concentration were plotted for two cases considered above; firstly, when IW phase velocities and wind drift were different and kinematic modulation dominated over the film effect, and secondly, when IW and drifting films were in resonance, resulting in strong modulations of surfactant concentration and in enhanced "film modulation" mechanisms. Figure 15 illustrates that the RMTF calculated according to the model was consistent with the measurements. Similar to radar contrasts, the modulations in our experiments were determined by strong ("cascade") modulation of the nonlinear structures generated on the profile of steep cm–dm-scale GCW.

Now, we briefly discuss the results of field observations of radar backscatter variations over IWs presented in Section 3. The IW phase velocities estimated from radar panoramas of slicks and ADCP measurements were about 27–28 cm/s. These values are approximately consistent with the theoretical values of about 29 cm/s obtained from the dispersion relationships for IWs in a two-layer fluid (see [39]):

$$C(k) = \sqrt{\delta\rho/\rho_0 g \cdot \kappa^{-1} \cdot [cth(\kappa h) + cth(\kappa H - h)]^{-1}}, \tag{24}$$

where $\delta\rho/\rho_0$ is the relative density difference between the upper and low water layers with thicknesses $h$ and $H - h$, respectively. The wind was directed towards the shore, and we estimated the wind drift velocity to be 3% of the wind speed, which was about

12 cm/s. When summing the wind drift and the mean current of 17 cm/s measured in the upper layer at a depth of 1 m, one comes to the value of the total mean surface velocity of 29 cm/s. The measured and theoretical IW velocities were consistent with each other; therefore, one can suppose that the velocities in the bulk water are rather small and do not alter the IW phase velocity. The projection of the surface velocity on the IW propagation direction is close to the IW phase velocity. Hence, according to the theory (see, Section 2) surfactants should be concentrated over the troughs/rear slopes of IWs; this was supported by our observations. Strong reductions in Ka-band radar intensity in slicks, of about 10–15 dB (see, Figure 12), indicate that surfactants are effectively accumulated in the slick bands and strongly suppress short wind waves. We suppose that radar backscattering is determined by capillary ripples with Bragg wavelengths of about 0.5 cm, and these ripples are, in general, both free waves and bound PCR. The latter significantly contribute to the spectrum of capillary waves, very likely dominating over free wind ripples because the latter are ineffectively generated at low wind speeds and particularly in film-covered areas. Then, the proposed cascade mechanism of strong modulations of PCR associated with accumulation of surface films by IW can be effective, and this conclusion is consistent, at least qualitatively, with the observed strong suppression of radar return in the slick bands.

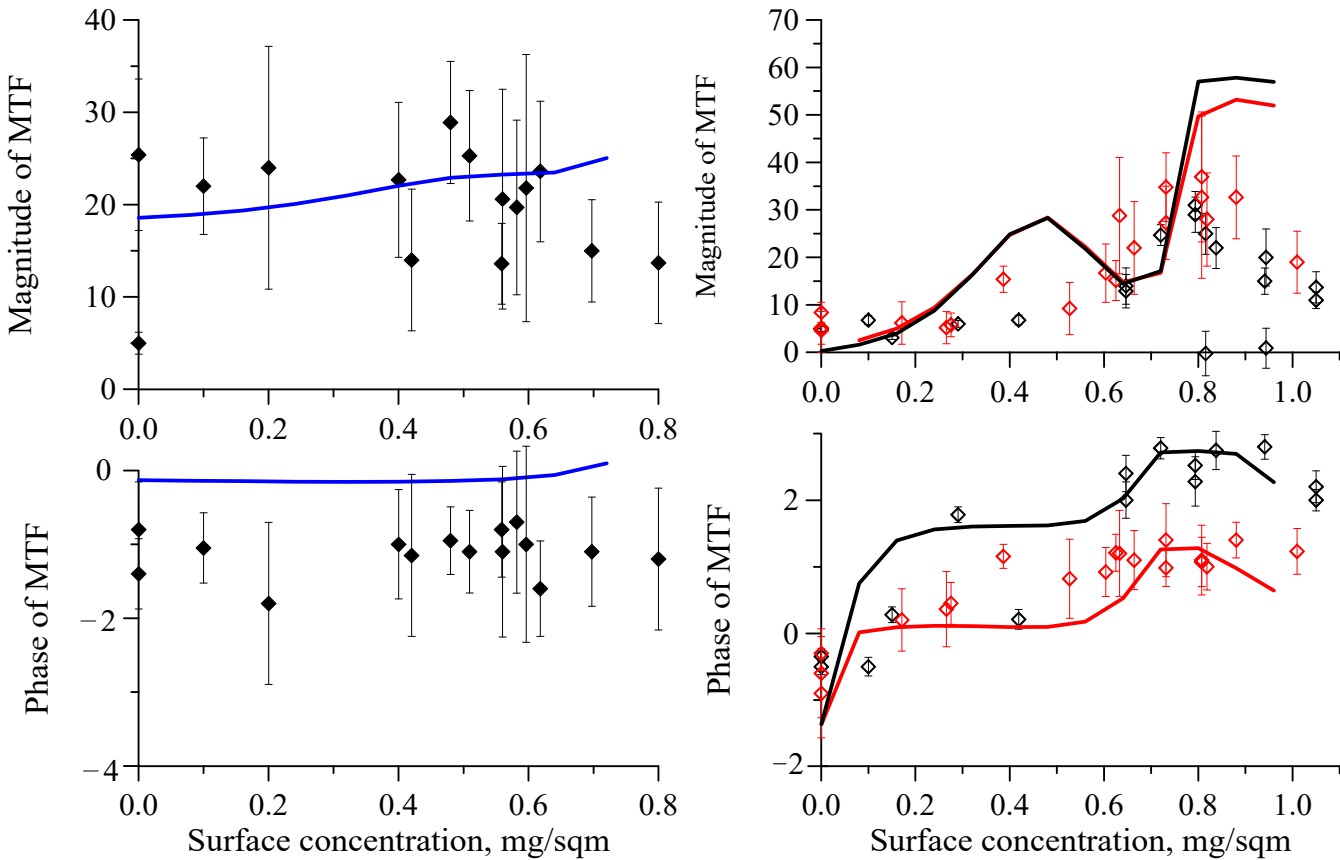

**Figure 15.** RMTF magnitude and phase for non-resonance (**left**) and resonance (**right**) calculated according to model (curves) and experimental data (symbols).

More detailed descriptions and analyses of the field experiments will be given elsewhere.

## 6. Conclusions

- A phenomenological model of the modulation of short wind waves and Ka-band radar return due to internal waves in the presence of surfactant films has been elaborated. Wave tank studies on the IW modulation of surfactant concentration and of radar

backscatter in the field of IW orbital currents were carried out for two different cases. In the first case, the IW phase velocity was large compared to the velocity of surfactant films due to wind drift; in the second case, these velocities were close to each other—non-resonance and resonance cases, respectively. It has been shown theoretically and in experiments that surfactants at resonance were strongly comparable to non-resonance cases concentrated over the IW troughs or the rear slopes of IW;

- It has been observed that Ka-band radar backscatter was reduced when a film was applied on water and the radar contrast increased with surfactant concentration and, in practice, did not depend on whether resonance on non-resonance conditions were realized;

- Modulation of radar return (the radar modulation transfer function) practically did not depend on surfactant concentration for non-resonance case, but strongly increased with concentration at resonance. The results were explained by different modulation mechanisms for these two cases, i.e., the straining modulation of wind waves due to IWs was a dominant mechanism for non-resonance cases, whereas the modulation of radar return at resonance was determined mostly by the strong modulation of surfactants. In both cases, the radar backscatter was modulated due to variations of parasitic capillary ripples generated by steep cm–dm-scale waves (cascade modulation), rather than of free Bragg capillary mm-scale waves. Some field observations have been presented, which qualitatively confirmed the effect of strong modulations of Ka-band radar backscatter due to IWs in the presence of resonance drift of surfactant films;

- It has been shown that the developed modulation model was consistent with experimental results;

- The mechanism of strong (cascade) modulation of microwave Ka-band radar backscatter studied in this paper is very important for correct interpretations of IW signatures in radar imagery of the sea surface. The quasi-linear models of hydrodynamic straining modulation combined with Bragg theory underestimate the IW radar signature contrasts, whereas the proposed cascade model, taking into account the strong nonlinearity of short wind waves, gives more realistic values of the contrasts and the RMTF. We plan to extend the model in the future to a wider range of wind wave spectra in order to explain IW signatures in satellite imagery in X/S radar bands.

**Author Contributions:** Conceptualization, S.A.E.; methodology, S.A.E. and I.A.S.; investigation, I.A.S. and I.A.K.; data curation, I.A.S. and I.A.K.; writing, S.A.E. All authors have read and agreed to the published version of the manuscript.

**Funding:** This work was performed within the State Assignment (project no. 0729-2020-0037).

**Conflicts of Interest:** The authors declare no conflict of interest.

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
