# Peer review of "Strong Modulation of Short Wind Waves and Ka-Band Radar Return Due to Internal Waves in the Presence of Surface Films. Theory and Experiment"

_remotesensing, doi:10.3390/rs13132462_

Round 1
Reviewer 1 Report
The manuscript "Strong modulation of short wind waves and Ka-band radar re-turn due to Internal Waves in the presence of surface films. Theory and Experiment" is a combination of theory and laboratory experiment. It is well written and clearly articulated. I would like to make two small remarks: Figures 5, 6 and others do not decipher what the different colors mean. There is no explanation in the text of the article either.The discussion section very briefly describes a field experiment in the Black Sea. I would like the authors to compare the results of laboratory experiments, to which the article is primarily devoted to the results of the field experiment mentioned in passing. Otherwise, it makes no sense to mention the field experiment.
Author Response
The manuscript "Strong modulation of short wind waves and Ka-band radar re-turn due to Internal Waves in the presence of surface films. Theory and Experiment" is a combination of theory and laboratory experiment. It is well written and clearly articulated.
Q: I would like to make two small remarks: Figures 5, 6 and others do not decipher what the different colors mean. There is no explanation in the text of the article either.
A: The captions to figures 5,6 explain the curves in different colors. Corrected.
Q: The discussion section very briefly describes a field experiment in the Black Sea. I would like the authors to compare the results of laboratory experiments, to which the article is primarily devoted to the results of the field experiment mentioned in passing. Otherwise, it makes no sense to mention the field experiment.
A: We have extended the description of the field experiment, see, Sections 3 (experiment), Section 4 (results), Section 5 (discussion).
Reviewer 2 Report
Starting from my limited knowledge of this area, the article seems extremely detailed to me. It is structured and with a very strong physical / mathematical basis. I suggest accepting the article after reviewing English by a native speaker and adding some future applications and future development/applications
Author Response
Q: Starting from my limited knowledge of this area, the article seems extremely detailed to me. It is structured and with a very strong physical / mathematical basis. I suggest accepting the article after reviewing English by a native speaker and adding some future applications and future development/applications
A: The authors have revised the English writing of the paper. Some future development/applications are mentioned in Section 6 (Conclusions).
Reviewer 3 Report
Review on “Strong modulation of short wind waves and Ka-band radar return due to Internal Waves in the presence of surface films. Theory and Experiment”
This paper studies the modulation of wind waves and Ka-band radar return due to internal waves in the presence of surface films, and several conclusions are drawn. However, the organization of the paper may be improved, e.g., the wave tank data are briefly analyzed in Section 4, and then some data and result are given in the section of “5. Discussion”. Moreover, the English writing are difficult to be understood. Specific comments are as follows.
The title says “strong modulation of short wind waves and Ka-band radar return due to Internal Waves in the presence of surface films”, so the influence of surface film on short wind wave spectrum should be important. However, it is only studied in Section 2 based on previous models, while it is seldom studied in the analysis of result by wave tank data (Figure 8). So the more study on the influence of surfactant on short wind wave should be given, by comparing the theory and experimental data.
Equation (1): All the variables in the equations should be clearly defined. The same problems also exist in other equations.
Figure 1: What does the arrow mean? It should be clearly stated. The same problems also exist in other figures.
In section 3, more information on the experiment should be given.The aim of the paper is to study the Ka-band radar return, but little information on the radar was illustrated (Line 337), and it is difficult to evaluate the radar data. For example, the main configurations of the radar, which are important for the detection of different waves and the estimation of Doppler spectra. The polarization is changed by “simply turning the radar antenna around its axis” (section 5), is there any influence on the working of the radar in this way?
Figure 2: More parameters on the wave tank may be labeled, such as the depth of pycnocline, the height of antenna, the incline angle of radar antenna, etc.
Line 377: “and then RMTF was calculated according to (2)”. It should be Equation (20)?
Lines 387-400: What are the IW wavelength and period in the first experiment? And what’s the amplitude of the IW in the second experiment?
Figures 5 & 6: legends should be added to show the meanings of different lines.
Section 4.2.1: The paper is to study the modulation of short wind waves by surface films, so the wind wave spectra in Figure 8 should be compared with that predicted by the model in Section 2, by using the same conditions as those.
Section 5: The field observation is important to show the validity of the model, it is confusing to show the field observation briefly here, and it may be better to improve it.
Lines 556-558: “Strong reductions of Ka-band radar intensity (10-15 dB, see, Figure 14) were recorded in the film areas located over IW troughs in the leading part of IW train and shifted towards the forward slope for the rear part”
Where are the rear, the leading part of IW train and the forward slope in Figure 14?
The abscissa of Figure 14 is “Elapsed time”, what does “IW propagation direction is from right to left” mean?
Line 564: “5. Conclusions” should be section 6.
There are many long sentences throughout the paper (e.g., Lines 566-572), which are difficult to understand, so it will be better to revise the English writing of the paper.
Author Response
This paper studies the modulation of wind waves and Ka-band radar return due to internal waves in the presence of surface films, and several conclusions are drawn. However, the organization of the paper may be improved, e.g., the wave tank data are briefly analyzed in Section 4, and then some data and result are given in the section of “5. Discussion”. Moreover, the English writing are difficult to be understood. Specific comments are as follows.
The title says “strong modulation of short wind waves and Ka-band radar return due to Internal Waves in the presence of surface films”, so the influence of surface film on short wind wave spectrum should be important. However, it is only studied in Section 2 based on previous models, while it is seldom studied in the analysis of result by wave tank data (Figure 8). So the more study on the influence of surfactant on short wind wave should be given, by comparing the theory and experimental data.
We improved the paper structure, in particular , we extended the field experiment description in Section 3, added field results in Section 4 and discussion in Section 5.
The influence of surfactants on short wind waves was studied in detail in our previous papers, and we included an additional reference on the most recent and advanced paper on this problem (see, ref.[38]). We also added some sentences about the effect of films on wind waves (original Figure 8, now it is Figure 9) and estimated the damping effect (contrast) both in experiment and according to our model and found them consistent to each other (see, paragraph after Fig 9).
Q: Equation (1): All the variables in the equations should be clearly defined. The same problems also exist in other equations.
A: Corrected. We have defined all the variables in the paper. We introduced an upper tilde denoting the functions varying due to non uniform currents, those with a subscript “0” corresponding to undisturbed, steady state values.
Q: Figure 1: What does the arrow mean? It should be clearly stated. The same problems also exist in other figures.
A: Corrected. The arrow shows the direction of propagation of IW.
Q: In section 3, more information on the experiment should be given.The aim of the paper is to study the Ka-band radar return, but little information on the radar was illustrated (Line 337), and it is difficult to evaluate the radar data. For example, the main configurations of the radar, which are important for the detection of different waves and the estimation of Doppler spectra. The polarization is changed by “simply turning the radar antenna around its axis” (section 5), is there any influence on the working of the radar in this way?
A: Additional information about the radar is inserted in the paper. See, page10 in the revised paper, the second paragraph after figure 3.
Q: Figure 2: More parameters on the wave tank may be labeled, such as the depth of pycnocline, the height of antenna, the incline angle of radar antenna, etc.
A: The depth of pycnocline, the height of antenna, the incline angle of radar antenna have been identified in Section 3 (lines 270, 339, 340 of the old pdf –version)
Q: Line 377: “and then RMTF was calculated according to (2)”. It should be Equation (20)?
A:Corrected
Q: Lines 387-400: What are the IW wavelength and period in the first experiment? And what’s the amplitude of the IW in the second experiment?
A: Corrected, see the captions to Figures 5,6.
Q: Figures 5 & 6: legends should be added to show the meanings of different lines.
- Corrected. The captions to figures 5,6 explain the curves in different colors.
Q: Section 4.2.1: The paper is to study the modulation of short wind waves by surface films, so the wind wave spectra in Figure 8 should be compared with that predicted by the model in Section 2, by using the same conditions as those.
- According to theory wind waves in the cm-dm-wavelength range of the spectrum are suppressed in about 1.5- 2 times that is approximately consistent with experiment.
Q: Section 5: The field observation is important to show the validity of the model, it is confusing to show the field observation briefly here, and it may be better to improve it.
A: We have extended the description of the field experiment. Information about the field experiment is entered in the sections Experiment (3), Results (4) and Discussion (5). All comments of the reviewer (see below) concerning the description of the field experiment are taken into account.
Lines 556-558: “Strong reductions of Ka-band radar intensity (10-15 dB, see, Figure 14) were recorded in the film areas located over IW troughs in the leading part of IW train and shifted towards the forward slope for the rear part”
Where are the rear, the leading part of IW train and the forward slope in Figure 14?
The abscissa of Figure 14 is “Elapsed time”, what does “IW propagation direction is from right to left” mean?
Q: Line 564: “5. Conclusions” should be section 6.
A: Corrected
Q: There are many long sentences throughout the paper (e.g., Lines 566-572), which are difficult to understand, so it will be better to revise the English writing of the paper.
A: The authors have revised the English writing of the paper.